

**Mercury distribution and transport in the North Atlantic Ocean along**
**the GEOTRACES-GA01 transect**
Daniel Cossa[1], Lars-Eric Heimbürger[2], Fiz F. Pérez[3], Maribel I. García-Ibáñez[3], Jeroen
E. Sonke[4], Hélène Planquette [5], Pascale Lherminier[6], Géraldine Sarthou[5]
[1]*ISTerre, Université Grenoble Alpes, CS 40700, F-38058 Grenoble Cedex 9, France*
[2]*Aix Marseille Université, CNRS/INSU, Université de Toulon, IRD, Mediterranean Institute of*
*Oceanography (MIO) UM 110, Marseille, France*
[3]*Instituto de Investigaciones Marinas, CSIC, Eduardo Cabello 6, E-36208 Vigo, Spain*
[4]*CNRS, GET-OMP, 14 Ave. E. Belin, F-31240 Toulouse, France*
[5]*LEMAR, Université de Bretagne Occidentale, F-29280 Plouzané, France*
[6]*IFREMER, Brittany Center, LPO, BP 70, F-29280 Plouzané, France*
***Abstract***
We report here the results of total mercury (HgT) determinations along the 2014
GEOTRACES GEOVIDE cruise (GA01 transect) in the North Atlantic Ocean (NA) from
Lisbon (Portugal) to the Labrador coast (Canada). Ninety-seven percent of the HgT
concentrations of unfiltered samples ($HgT_{UNF}$) ranged between 0.16 and 1.00 pmol L$^{-1}$.
The geometric mean was 0.51 pmol L$^{-1}$ for the 535 samples analysed. The dissolved
fraction (< 0.45 µm) of HgT, determined on 141 samples, averaged 78 % of the $HgT_{UNF}$
for the whole water column. $HgT_{UNF}$ concentrations increased eastwards and with depth
from Greenland to Europe and from sub-surface to bottom waters, respectively. The Hg
distribution mirrored that of dissolved oxygen concentration, with highest $HgT_{UNF}$
levels associated with oxygen-depleted zones. The statistically significant ($p < 0.01$)
relationship between $HgT_F$ (filtered samples) and the apparent oxygen utilization
confirms the nutrient-like behavior of Hg in the NA. An extended Optimum
Multiparameter Analysis allowed us to characterize $HgT_{UNF}$ concentrations in the
different Source Water Types (SWTs) present along the transect. Mean $HgT_{UNF}$
concentrations ranged from $0.26 \pm 0.03$ pmol L$^{-1}$ in the Irminger Subpolar Mode Water



to $1.02 \pm 0.02$ pmol $L^{-1}$ in the lower North East Atlantic Deep Water. Anthropogenic
Hg-enriched SWTs were found in the upper oceanic layers (i.e., East North Atlantic
Central Water and Subarctic Intermediate Water). The change in anthropogenic Hg
concentrations in the Labrador Sea Water, during its eastward journey, suggests a
continuous decrease of Hg content in this water mass over the last decades. Calculation
of the water transport driven by the Atlantic Meridional Overturning Circulation across
the Portugal-Greenland transect indicates a northward Hg transport within the upper
limb, and a southward Hg transport within the lower limb, with a resulting net transport
from low latitudes to the polar zones of about 111 kmol $yr^{-1}$.

**1.    Introduction**
Many trace elements have a physiological function for marine phytoplankton (e.g.,
Morel et al., 2003). Others, such as mercury (Hg), have no known beneficial role for
most of biological systems, and may even provoke toxicological disturbances of marine
ecosystems (e.g., Fitzgerald et al., 2007; Mason et al., 2012). The global Hg
biogeochemical cycle is dominated by the atmosphere-ocean exchanges, with
atmospheric Hg deposition representing the principal source of inorganic Hg to the open
ocean (e.g., Mason et al., 2012; Sonke et al., 2013). Once deposited at the ocean surface,
Hg penetrates the ocean interior both *via* water mass formation, i.e., sinking of surface
waters to depth (the solubility pump), and *via* Hg sorption, sinking, and subsequent
remineralization of biogenic particles produced in the euphotic zone (the biological
pump) at depth. Anthropogenic Hg emissions to the atmosphere have severely altered
the Hg cycle during the last centuries (e.g., Fitzgerald et al., 2007; Lamborg et al.,

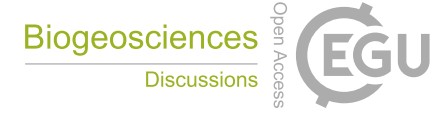

2014). Modern Hg concentrations in the global atmosphere are more than three times
the pre-industrial Hg concentrations, leading to a doubling of the Hg concentrations in
surface ocean (Mason et al., 2012; Lamborg et al., 2014; Amos et al., 2015).
Recognizing the toxicological potential of this Hg anthropization of the oceans, national
and international agencies have proposed and enforced regulations since the 1980s to
reduce man-made Hg emissions to the environment (Pirrone and Mahaffey, 2005). For
this purpose, United Nation Environment Program led to the adoption of the Minamata
Convention (UNEP, 2013). Within this context, it is important to characterize the Hg
content of various water masses of the Global Ocean and to monitor their temporal
variations.
The North Atlantic Ocean (NA) is one of the places where deep water formation
(e.g., Labrador Sea Water, LSW) is particularly important and, thus, where atmospheric
Hg deposition can enter into the ocean interior *via* the solubility pump, offering a
unique opportunity for studying the oceanic response to changes in atmospheric Hg
inputs. In addition, recent improvements in the precision and accuracy of total mercury
(HgT) analyses allowed a more detailed description of the NA HgT distribution,
especially within the GEOTRACES program (e.g., Cossa et al., 2011; Lamborg et al.,
2014; Bowman et al., 2015; Munson et al., 2015). New GEOTRACES results provide
evidence that the main features of the HgT distribution patterns are related to the
physical and biogeochemical oceanographic properties of waters masses. In the NA, the
GEOTRACES-GA03 zonal and meridional transects, sampled in 2010 and 2011, covered
the NA from East to West between 18°N and 40°N, from Africa to USA coasts. Here,
we report the results of the GEOVIDE cruise, along the GEOTRACES-GA01 transect,



which targeted the NA from 40°N to 60°N, from Portugal to Newfoundland, *via* the
southern tip of Greenland (Fig. 1).

This article provides (i) a high-resolution description of the HgT distribution in

the waters of Subpolar and Subtropical gyres of the NA, (ii) a characterization the HgT
concentrations of the main water masses of the NA, (iii) an estimate of the temporal
change of anthropogenic Hg in LSW, and (iv) a quantification the HgT transport
associated with the upper and lower limbs of the Atlantic Meridional Overturning
Circulation (AMOC).

**2.    Oceanographic context**
A full description of the water masses along the GEOTRACES-GA01 transect can be
found in García-Ibáñez et al. (this issue). Briefly, the North Atlantic Current (NAC)
conveys the warm salty surface waters from subtropical regions northwards to the
subpolar regions, where they are cooled down by heat exchange with the atmosphere
(Fig. 1). The intermediate and deep waters formed this way fill up the Global Ocean,
initiating the southward-flowing limb of the AMOC (e.g., McCartney and Talley, 1984;
Lherminier et al., 2010). In addition, the general circulation pattern is characterized by
the subtropical and the subpolar gyres (Fig. 1).

In the Subtropical Gyre (Fig. 1), several water masses are stacked up from surface

to bottom: (i) the mixed layer, (ii) the East North Atlantic Central Water (ENACW),
(iii) the Mediterranean Water (MW), (iv) the Labrador Sea Water (LSW) and (v) the
lower North East Atlantic Deep Water (NEADW$_L$), which contains about 30%





Antarctic Bottom Water (AABW) (García-Ibáñez et al., 2015). The transformation of
ENACW leads to the formation of different mode waters including the Subpolar Mode
Waters (SPMWs) (McCartney and Talley, 1982; Tsuchiya et al., 1992; van Aken and
Becker, 1996; Brambilla and Talley, 2008; Cianca et al., 2009). SPMWs are the near-
surface water masses of the Subpolar Gyre of the NA characterized by thick layers of
nearly uniform temperature, often defined as SPMW with a temperature as a subscript
($SPMW_8$ for example). SPMWs are formed during winter convection at high latitude,
due to atmospheric freshening of surface waters originating from the subtropical gyre
(McCartney, 1992). SPMWs participate in the upper limb of the AMOC and provide
much of the water that is eventually transformed into the several components of North
Atlantic Deep Water (NADW; Brambilla and Talley, 2008).

In the Subpolar Gyre, ocean-atmosphere interaction is particularly intense. The

cooling down of subtropical waters produces dense waters, triggering the deepening of
the mixed layer and further leading to deep convection. The main NA convection zones
are located in the Labrador (LS), Irminger (IrS) and Nordic seas (NS) (Fig. 1).
Convection in those zones leads to the formation of intermediate and deep waters such
as LSW, Denmark Strait Overflow Water (DSOW) and Iceland-Scotland Overflow
Water (ISOW). LSW and ISOW are the main components of NEADW, and the all three
are the components of the NADW, which constitutes the cold deep southward-flowing
limb of the AMOC, flowing towards the Southern Ocean in the western Atlantic basin.
LSW has been variably produced in the past fifty years depending on the intensity of
winter convection, linked to the intensity of the North Atlantic Oscillation (e.g., Rhein
et al., 2002; Cianca et al., 2009; Yashayaev and Loder, 2016). Depths of winter





convection in the LS vary from a few hundred meters (the early 2000s) to over 2000 m
(early 1990s). The LSW is a thick layer in the LS and thins out as it travels
southwestwardly. It spreads out into the entire NA, filling the Subpolar Gyre and
entering the Subtropical Gyre. Within the Subpolar Gyre, LSW is marked by a salinity
minimum above the ISOW. In both gyres, the well-ventilated LSW is noticeable by a
marked oxygen maximum.
In the deep convection zones, atmospheric compounds, including those of volatile
chemicals such as chlorofluorocarbons (CFCs) (Smethie et al., 2000) are injected into
the ocean interior. In the Eastern NA, very low CFC concentrations below LSW
characterize the NEADW$_L$ and attest of negligible atmospheric exchange (McCartney
and Talley, 1984; Rhein et al., 2002). Hg partly follows the pathway of CFCs; it
intrudes into the interior of the Atlantic Ocean through surface waters at higher latitudes
in the NA and is conveyed by the AMOC. The upper part of LSW and DSOW have
relatively high CFC concentrations (Azetsu-Scott et al., 2003). The CFC maximum is
located between 1200 -1500 m-depth for LSW but can penetrate deeper down to 2000
m-depth, and the CFC maximum for DSOW is at 3500 m (Smethie et al., 2000). Hg
enters the Ocean's interior by a similar mechanism, and, in productive zones, Hg can
also enter with sinking organic particles.

**3.    Material and methods**
*3.1.   Sampling*



Water samples were collected during the French-led GEOVIDE cruise (GEOTRACES-
GA01 transect), on board the RV *Pourquoi Pas?* sailing from Lisbon (Portugal) on May
15$^{th}$ to arrive on June the 30$^{th}$ 2014 in St John's (Newfoundland, Canada) (Fig. 1).
Seventy-eight (78) stations (Table S1) were occupied for hydrographic profiles (CTD,
dissolved oxygen, nutrients, etc.), among which 29 included trace metal sampling.
Sampling and water treatment for HgT determination (Lamborg et al., 2012; Cutter et
al., 2017) were performed using ultra-trace techniques following the GEOTRACES
recommendations. During the GEOVIDE cruise, an epoxy-coated aluminum rosette,
equipped with 12 L GO-FLO (General Oceanics®) bottles initially cleaned following the
GEOTRACES procedures (Cutter and Bruland, 2012), was deployed on a 6 mm Kevlar
hydrowire. The rosette was also equipped with probes for pressure, conductivity,
temperature, dissolved oxygen, fluorometry and transmission measurements (titanium
SBE model 911-plus, Sea-Bird Electronics®). Specifically, for Hg determination, all
material in contact with the seawater samples was made of Teflon or Teflon coated, acid
cleaned and rinsed with ultrapure water (Milli-Q, Millipore®) prior to utilization.
Original vent fixture and sampling valves of the GO-FLO bottles were replaced with
Teflon (PTFE) ones. GO-FLO bottles were sub-sampled under a laminar flow bench
inside a trace metal clean container. The efficiency of the High-Efficiency Particulate
Air filter (HEPA, 0.3 µm) in the container was checked with a Coulter Counter during
the cruise. All subsequent sample treatments (including filtration) and Hg analyses were
also performed in class 100 clean containers. For sample filtrations, acid-washed 0.45
µm polycarbonate membranes (Nuclepore) were preferred to cellulose acetate or
polyethersulfone membranes proposed in the GEOTRACES protocols (Fig. S1). Sub-



samples were stored in Teflon bottles (FEP) until the on board HgT analyses, which
occurred within 6 hours after sampling.
### *3.2.   Chemical analyses*
In order to access all Hg species, the release of Hg from its ligands was achieved by a
BrCl solution (50 µL of a 0.2 N solution is added to a 40-mL sample), and then the Hg
was reduced with an acidic $SnCl_2$ solution (100 µL of a 1 M solution is added to a 40-
mL sample). Potassium bromide (Sigma Aldrich, USA) and potassium bromate (Sigma
Aldrich, USA) were heated for 4 h at 250°C to remove Hg traces before making up
BrCl solution with freshly double-distilled HCl (Heimbürger et al., 2015). The
generated Hg vapor was amalgamated on a gold trap and then released by heating into
an atomic fluorescence spectrometer (AFS). We used two AFS systems in parallel
(Tekran® Model 2500, Brooks® Model 3), both calibrated against the NIST 3133
certified reference material. This technique, initially described by Bloom and Crecelius
(1983) and subsequently improved by Gill and Fitzgerald (1985), is now an
authoritative procedure officialised by the US-EPA as method 1631 (EPA, 2002). The
definitions of detection limit (DL), reproducibility and accuracy given here are adopted
from Taylor (1987) and Hewitt (1989). Using a mirrored quartz cuvette (Hellma®)
allowed for an "absolute DL", defined as two times the electronic noise magnitude, as
low as 1.7 femtomoles. However, in practice for trace measurements, the DL is
governed by the reproducibility of the blank values, and calculated as 3.3 times the
standard deviation of blank values. The blank was determined on a purged Hg-free
seawater sample spiked with reagents (i.e., BrCl and $SnCl_2$). The mean (± standard
deviation) of blanks measured during the GEOVIDE cruise was 3.2 ± 1.0 femtomoles.





Thus, for a 40-mL seawater aliquot, the DL expressed in HgT concentration was 0.07
pmol L$^{-1}$. The reproducibility (coefficient of variation of six replicate measures) varied
according to the concentration level between 5 and 15 %. The accuracy of HgT
measurements was tested using ORMS-5 certified reference material (CRM) from the
National Research Council of Canada (http://www.nrc-cnrc.gc.ca/), as spike addition to
a purged Hg-free seawater sample. Measurements were always within the given
confidence interval. To ensure good data quality, and as a continuity of previous efforts
(Cossa and Courau, 1990; Lamborg et al., 2012), we organized the 2014 GEOTRACES
intercalibration exercise for total HgT and methyl Hg as a part of the GEOVIDE cruise.
The intercalibration sample was taken on June 22$^{nd}$, 2014 in the LS at 49.093°W,
55.842°N, and 2365 m-depth. The sample was sent out to 10 participating laboratories.
This station was also planned as crossover station within the 2015 Arctic GEOTRACES
effort (Canadian cruise) but has been changed subsequently to another location. Our
results compare well with the consensus values, HgT = 0.63 ± 0.12 pmol L$^{-1}$, n = 8. We
measured the 2014 GEOTRACES intercalibration sample twice for HgT and obtained 0.51
(22$^{nd}$ June 2014, on board) and 0.58 pmol L$^{-1}$ (30$^{th}$ October 2014, home lab).
*3.3.   Extended Optimum Multiparameter analysis*
We used an extended Optimum Multiparameter (eOMP) analysis to characterize the
water mass HgT$_{UNF}$ concentrations along the GEOTRACES-GA01 transect (García-Ibáñez
et al., 2015, this issue). The eOMP analysis quantifies the proportions of the different
Source Water Types (SWTs) that contribute to a given water sample. The HgT$_{UNF}$
concentration of each SWT, [HgT$_{UNF}$]$_i$, was estimated through an inversion of the SWT
fractions given by the eOMP analysis. Such an approach was successfully applied to



dissolved-organic-carbon water mass definitions in the NA (Fontela et al., 2016) and for
evaluating the impact of water mass mixing and remineralization on the $N_2O$
distribution in the NA (de la Paz et al., 2017). Here, we performed an inversion of a
system of 430 equations ($HgT_{UNF}$ samples) and 11 unknowns ($[HgT_{UNF}]_i$). Samples for
which the difference between the observed $HgT_{UNF}$ and the predicted $HgT_{UNF}$ values by
the multiple linear regression (Eq. 1 below) was three times greater than the standard
deviation were removed from the analysis. Nine samples were concerned: Sta. 2 (125
m), Sta. 11 (793 m), Sta. 11 (5242 m), Sta. 13 (1186 m), Sta. 15 (170 m), Sta. 19 (99
m), Sta. 26 (97 m), Sta. 32 (596 m), and Sta. 38 (297 m). The SWTs were characterized
by potential temperature, salinity, and macronutrients. The eOMP was restricted to
depths below 75 m in order to avoid air-sea interaction effects. The eOMP gave us the
fractions of the 11 SWTs, and we resolved the following expression to estimate the
$[HgT_{UNF}]_i$:
$$[HgT_{UNF}]_j = \sum_{i=1}^{11} SWT_i^j * [HgT_{UNF}]_i + \varepsilon_j \quad (j = 1...430) \quad (1)$$
where $[HgT_{UNF}]_j$ represents the measured $HgT_{UNF}$ concentration for each sample *"j"*,
$SWT_i^j$ the proportion of SWT *"i"* to sample *"j"* (obtained through the eOMP),
$[HgT_{UNF}]_i$ the $HgT_{UNF}$ concentration for each SWT *"i"* (unknow), and $\varepsilon_j$ the residual.
The 430 $\varepsilon_j$s of the inversion presented a null mean and a standard deviation of 0.085
pmol $L^{-1}$ (R = 0.84).
**3.4.  *Mercury transport calculation***
Velocity fields across the GEOTRACES-GA01 transect were calculated using inverse
model constrained by Doppler current profiler velocity measurements (Zunino et al.,



this issue) an overall mass balance of $1 \pm 3$ Sv to the North (Lherminier et al., 2007,
2010). The volume transport per SWT was computed by combining these velocity fields
with the results of the eOMP (García-Ibáñez et al., this issue). Finally, the HgT$_{UNF}$
transports per water mass were calculated through Eq. (2):

4.  $T_{HgT_{\mathrm{UNF}}} = \sum_{i=1}^{11} T_{SWT_i} * [HgT_{\mathrm{UNF}}]_i * \rho_i$         (2)

where $T_{SWTi}$ is the volume transport of SWT "$i$", $[HgT_{UNF}]_i$ is the HgT$_{UNF}$ concentration
for each SWT "$i$" (from Eq. 1), and $\rho_i$ is the density of the SWT "$i$".
The inverse model configuration for the GEOVIDE cruise data is described in Zunino et
al. (this issue). The inverse model is based on the least-squares formalism, which
provides errors on the velocities and associated quantities such as the magnitude of the
AMOC (estimated in density coordinate) and the heat flux (Lherminier et al., 2010).
The inverse model computes the absolute geostrophic transports orthogonal to the
section. The Ekman transport is deduced from the wind fields averaged over the cruise
period and added homogeneously in the upper 40 m (Mercier et al., 2015). The transport
estimates of the inverse model across the section have been validated by favorable
comparisons with independent measurements (Gourcuff et al., 2011; Daniault et al.,
2011; Mercier et al., 2015).

**5.    Results**
HgT$_{UNF}$ concentrations along the GEOTRACES-GA01 transect ranged from 0.16 to 1.54
pmol L$^{-1}$ (n = 535), these data being log-normally distributed, positively skewed
(Skewness = 1.1; Kurtosis = 2.1; Fig. S2) and with 97 % of the values lower than 1.00



pmol L$^{-1}$. The geometric mean and the median were 0.51 pmol L$^{-1}$, whereas the
arithmetic mean and standard deviation were 0.54 and 0.19 pmol L$^{-1}$, respectively.
These concentrations are within the range found along the GEOTRACES-GA03 transect
(0.09–1.89 pmol L$^{-1}$, n = 605) that crossed the NA within the subtropical gyre from
18°N to 40°N (Bowman et al., 2015), but lower than the range and the unusually high
arithmetic mean determined in the South Atlantic along the GEOTRACES-GA10 transect
(0.39–3.39 pmol L$^{-1}$, n = 375; Bratkič, personal communication, and 1.45 ± 0.6 pmol L$^{-}$
$^{1}$; Bratkič et al., 2016, respectively).
The overall distribution of the HgT$_{UNF}$ concentrations along the GEOTRACES-
GA01 transect is represented in Fig. 2. The main feature of HgT$_{UNF}$ concentrations is an
eastward increase, from Greenland to Europe, and downward increase, from sub-surface
to bottom waters. In addition, highest and lowest (most variable) HgT$_{UNF}$ values were
encountered in surface/sub-surface waters, where Hg evasion to the atmosphere and
high particulate matter concentrations may generate low and high HgT$_{UNF}$
concentrations, respectively. Out of the 141 filtered samples that were analysed,
altogether, the filtered fraction of Hg (HgT$_{F}$) represents, on average, 78% (range: 36–
98%) of the HgT$_{UNF}$ (Fig. 3). Excluding the upper 100 m, where most of the particles
were present, the HgT$_{F}$ fraction represents, on average, 81% (range: 63–98%) of the
HgT$_{UNF}$. In the following sub-sections, detailed descriptions of the HgT$_{UNF}$ profiles for
the five following oceanographic environments are given: LS, IrS, Iceland basin (IcB),
Eastern North Atlantic basin (ENAB) and Iberian abyssal plain (IAP).
*5.1. Labrador Sea (Stas. 61 to 78)*



In the LS, the $HgT_{UNF}$ concentrations ranged from 0.25 to 0.67 pmol L$^{-1}$, with a mean of
0.44 ± 0.10 pmol L$^{-1}$ (n = 113, 1σ). Distribution, source, and cycling of Hg in the LS
have been described and discussed in detail in a companion paper (Cossa et al., in
press). In summary: high $HgT_{UNF}$ concentrations were found in the waters of the
Labrador Current (LC) receiving freshwaters from the Canadian Arctic Archipelago,
and in the waters over the Labrador shelf and rise. In the LSW formed during the 2014
winter convection, $HgT_{UNF}$ concentrations were low (0.38 ± 0.05 pmol L$^{-1}$, n = 23) and
increased gradually with depth (up to > 0.5 pmol L$^{-1}$) in the Northeast Atlantic Deep
Waters.
*5.2.  Irminger Sea (Stas. 40–60)*
$HgT_{UNF}$ concentrations in the IrS waters varied from 0.22 to 0.76 pmol L$^{-1}$, with a mean
of 0.45 ± 0.10 pmol L$^{-1}$ (n = 103). In the IrSPMW, which was encountered in the
upper1000 m near the East Greenland and the upper 500 m in the rest of the IrS (Fig. 4a
in García-Ibáñez et al., this issue), $HgT_{UNF}$ values span between 0.29 and 0.42 pmol L$^{-1}$
(Fig. 2). Deeper, $HgT_{UNF}$ increased up to 0.50 and 63 pmol L$^{-1}$ in LSW (~1000 m) and
ISOW (~2500 m), respectively. Lower $HgT_{UNF}$ concentrations (0.40–0.50 pmol L$^{-1}$)
were associated with DSOW in the very bottom waters (Sta. 42-44, Fig. 2).
*5.3.  Iceland Basin (Stas. 34–38)*
$HgT_{UNF}$ concentrations in the IcB ranged from 0.18 to 0.65 pmol L$^{-1}$, with a mean of
0.46 ± 0.10 pmol L$^{-1}$ (n = 51). In the top 100 m of the water column, $HgT_{UNF}$
concentrations were quite variable (0.25-0.62 pmol L$^{-1}$) probably as a result of the
counteracting importance of Hg evasion to the atmosphere and high particulate matter
concentrations. West of the IcB (Sta. 38), contrasting $HgT_{UNF}$ levels were found on both





sides at 500 m, characterized by a thermohaline gradient (Fig. 2a and b in García-Ibáñez
et al., this issue). In the top waters, $HgT_{UNF}$ levels were depleted to 0.18 pmol $L^{-1}$,
whereas, below 500 m, they were much higher and converge to values close to what we
found, at the same depths in the adjacent IrS (~0.60 pmol $L^{-1}$, Sta. 40). In the bottom
waters, constituted by more than 50% of ISOW (García-Ibáñez et al., this issue),
$HgT_{UNF}$ concentrations reached values > 0.50 pmol $L^{-1}$.

### *4.4.  Eastern North Atlantic Basin (Stas. 17–32)*

The $HgT_{UNF}$ concentrations in the ENAB varied from 0.18 to 1.14 pmol $L^{-1}$, with a
mean of 0.61 ± 0.18 pmol $L^{-1}$ (n = 174). The ENAB, also named Western European
Basin, is characterized by a complex vertical stratification of the water column, and the
presence of the Subarctic Front (SAF) that was located between Sta. 25 and 26 in the
GEOTRACES-GA01 transect (García-Ibáñez et al. and Zunino et al., this issue). West of
the SAF (Stas. 26-32), the mean $HgT_{UNF}$ concentrations was not statistically different
from that obtained for stations east of the SAF (Sta. 17-25): 0.63 ± 0.14 pmol $L^{-1}$ (n =
64) *versus* 0.59 ± 0.20 pmol $L^{-1}$ (n = 110). The $HgT_{UNF}$ vertical profiles at all the
stations of the ENAB were characterized by a complex but reproducible pattern
depicting (i) two maxima peaks (the upper at subsurface, the lower within the
intermediate waters), and below, (ii) a $HgT_{UNF}$ enhancement starting from 2500 m
towards the bottom (Fig. 2). The position and intensity of the peaks vary with longitude.
The upper peak, which occurs within the top 200 m, is only 0.48 pmol $L^{-1}$ at Sta. 29, but
reaches 1.14 pmol $L^{-1}$ at Sta. 19 (Fig. 2). The vertical position of maxima of the lower
peak and deepens eastwards, from 200 m down to 800 m, concurrently with an increase
of its amplitude (Fig. 2). The position of the upper peaks suggests a relation with the



abundance of phytoplankton, whereas the position of the lower peaks, which is close to
the maximum of Apparent Oxygen Utilization (AOU) that rose above 70 µmol L$^{-1}$ (Fig.
2), suggests a dependence on the organic matter remineralization (see Discussion
below). Between 1400 and 2500 m, in the layer corresponding to LSW, HgT$_{UNF}$
concentrations were quite uniform, with a mean concentration of $0.54 \pm 0.04$ pmol L$^{-1}$
(n = 18). HgT$_{UNF}$ concentration increased from 3000 m downwards to the sea bottom,
consisting of NEADW$_L$, where it reaches 0.95, 0.97, 1.03 and 1.13 pmol L$^{-1}$ at Sta. 21,
19, 25 and 23, respectively.
*4.5.    Iberian Abyssal Plain (Stas. 1–15)*
In the IAP, HgT$_{UNF}$ concentrations ranged from 0.19 to 1.54 pmol L$^{-1}$, with a mean of
$0.69 \pm 0.23$ pmol L$^{-1}$ (n = 94). The highest HgT$_{UNF}$ concentrations were measured in the
upper 100 m near the shelf slope. At Sta. 2, the only station on the European shelf
(bottom at 152 m), the HgT$_{UNF}$ concentrations increased from 10 m to the bottom, from
0.38 to 0.86 pmol L$^{-1}$, but did not differ from the open NA ocean levels. Off-shore, at
Sta. 1, 11, 13 and 15 (Fig. 2), the vertical distributions of HgT$_{UNF}$ presented a certain
similarity with those of the eastern ENAB, but with an additional third deep peak. As in
the eastern ENAB, the upper peak is associated with subsurface waters, and the second,
centered around 800 m, is associated with the oxygen minimum of SPMW$_8$. The third
peak, centered around 1100-1200 m, is associated with the salinity maximum of the
core of MW. The presence of a HgT$_{UNF}$ peak in the MW was still visible westwards, at
Sta. 17, 19 and 23, near 1100 m, as a shoulder of the main peak at 800 m (Fig. 2).
Deeper in the water column, HgT$_{UNF}$ increased gradually from 2000 m (LSW) to 3000

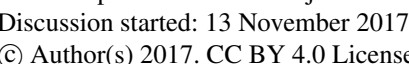



m (ISOW), 3500 m and below (NEADW$_L$), where HgT$_{UNF}$ concentrations reached 0.87
to 1.04 pmol L$^{-1}$ depending on the station.

**5.**      **Discussion**
*5.1.*    *Biogeochemical and hydrographical controls on HgT distribution*
Hg is dispersed in the atmosphere, deposited on sea surfaces, and drawn down to the
ocean interior with downwards convecting waters, and/or associated with sinking
particles. At depth, the dissolution of particulate matter, produced as a result of organic
matter microbiological remineralization, remobilizes Hg from particles produced in the
euphotic zone.

The biological pumping/regeneration process entails the existence of relationships

between Hg concentrations and nutrient or dissolved oxygen concentration (or AOU),
which are proxies of the organic matter remineralization (mainly the microbial
respiration) that the sample has experienced since it was last in contact with the
atmosphere. Such a biogeochemical behaviour, which is qualified of "nutrient-like"
behaviour, is observed in the present study (Fig. 4). The correlation coefficient (R)
between HgT$_F$ and the AOU, reached the highly statistically significant value of 0.87 (n
= 141, *p < 0.01*). Similar behaviour was already observed in the water column near the
shelf edge of the western European margin (Cossa et al., 2004), and elsewhere in the
NA (Lamborg et al., 2014; Bowman et al., 2015). Thus, biological uptake and
regenerative processes appear to control the oceanic Hg distribution in the Subpolar and
Subtropical gyres of the NA.





Hydrological circulation impacts the Hg distribution in the NA. We estimated the
$HgT_{UNF}$ (and AOU) values of each SWT using eOMP (Table 1). The correlation
coefficient between observed and predicted (eOMP-based) values through Eq. (1)
(Material and Methods section) for $HgT_{UNF}$ is 0.71. The estimated $HgT_{UNF}$
concentrations vary significantly between SWTs (ANOVA, *p < 0.01*), from $0.26 \pm 0.03$
to $1.02 \pm 0.02$ pmol $L^{-1}$, for the IrSPM to the $NEADW_L$, respectively. The low $HgT_{UNF}$
value in the IrSPMW, the youngest SPMW formed in the IrS as a result of air-sea
interaction of the waters transported northwards by the NAC (e.g., McCartney and
Talley, 1984), is similar to that found in the LSW formed during the 2014-winter
convection in the LS (Cossa et al., in press), consistently with hydrographic features,
which suggests that the IrSPMW is a precursor of LSW formed in the LS (Pickart et al.,
2003). The highest estimated $HgT_{UNF}$ concentration is calculated for $NEADW_L$, which
is the dominant water mass in the bottom IAP, with its main core below ~3500 m-depth
and spreading down to the bottom (García-Ibáñez et al., this issue), where it mixes with
the Hg-rich AABW, a deep-water mass originating from the Southern Ocean ($HgT_{AABW}$
$= 1.35 \pm 0.39$ pmol $L^{-1}$, Cossa et al., 2011). Thus, both AABW influence and organic
matter remineralization (AOU ~99 µmol $L^{-1}$, Table 1) converge to explain the Hg-
enrichment of $NEADW_L$. The same rationale can be drawn for the estimated $HgT_{UNF}$
concentration in MW ($0.75 \pm 0.04$ pmol $L^{-1}$, Table 1). MW originating from the
Mediterranean Sea is located just below $SPMW_8$, with a core at 1100 m (García-Ibáñez
et al., this issue). Both $SPMW_8$ and MW are characterized by elevated $HgT_{UNF}$
concentration and AOU values (see eOMP estimates in Table 1). Indeed, recent
measurements in the waters of the Western Mediterranean state $HgT_{UNF}$ values varying





between 0.53 and 1.25 pmol L$^{-1}$ within the layer that flows out of the Mediterranean Sea
at the Strait of Gibraltar (Cossa and Coquery, 2005; Cossa et al., 2017). In addition,
PIW and SPMW$_8$, present also relatively high HgT$_{UNF}$ and AOU concentrations (Table

1).

In order to sort out the influence of remineralization processes on the HgT$_{UNF}$ in

each SWT, we plotted the linear relationship of HgT$_{UNF}$ *versus* AOU, with the intercept
at zero (mineralization curve, Fig. 5). The departure of the estimated HgT$_{UNF}$
concentrations for each SWT from the remineralization curve shows that, among the
various SWTs, SAIW$_6$ and ENACW$_{12}$ exhibit Hg-enrichment, suggesting a significant
control by hydrographical features. These two SWTs are within the upper layer (the top
500 m), and thus more affected by atmospheric deposition. This observation suggests
that direct atmospheric deposition would be a significant source of Hg in the upper NA
waters, from the IrS to the ENAB. According to de Simone et al. (2016), anthropogenic
Hg emissions contribute 20−25 % to present-day Hg deposition, two-thirds of which is
deposited to the sea surface.

In summary, the distribution pattern of HgT$_{UNF}$ along the GEOTRACES-GA01

transect, modelled by mixing of SWTs (Fig. S3), stresses the importance of organic
matter regeneration and hydrological processes in Hg distribution in the NA.
*5.2.    Change in anthropogenic Hg in LSW*
Evidence for a decrease in the Hg anthropization in the NA waters can be obtained from
the comparison of the present results with those obtained twenty years ago with similar
clean sampling and analytical techniques. In a companion paper (Cossa et al., in press),
we have already compared the present findings for the convection layer in the LS with





the results of the 1993 International Oceanographic Commission cruise (Mason et al.,
1998). Between 1993 and 2014 the decrease in $HgT_{UNF}$ concentrations would have been
more than a factor of two ($1.14 \pm 0.36$ pmol $L^{-1}$ *versus* $0.40 \pm 0.07$ pmol $L^{-1}$). An
estimation of the anthropogenic Hg ($Hg_{Anth}$) concentrations in subsurface waters can be
inferred from the difference between measured $HgT_{UNF}$ concentrations and the
concentrations predicted based on a worldwide relationship between deep ocean Hg
concentrations and remineralised phosphate (Lamborg et al., 2014), with a Redfield
ratio of 141 between AOU and remineralized phosphate (Minster and Boulahdid, 1987)
a more representative value for the North Atlantic than the global value of 170 proposed
by Anderson and Sarmiento (1994). The LSWs take less than 20 years (Doney et al.,
1997) to flow eastward more than 3000 km from the LS eastward to the Subtropical
Gyre of the NA. Along its path, LSW bears the record of Hg solubility pumping at the
time of their formation, thus sampling along its flow path allows the observation of
decadal variations in anthropogenic Hg inputs to the NA. In the NA, estimation of
$Hg_{Anth}$ concentrations in the core of LSW, defined within potential density of 27.74 and
27.82, account for $36 \pm 0.07$ % of the $HgT_{UNF}$, and are one third lower for younger
waters (LS and IrS: $0.16 \pm 0.11$ pM, mean $\pm 1$ $\sigma$) than for older waters (IcB and ENAB:
$0.24 \pm 0.06$ pM, mean $\pm 1$ $\sigma$) (*t-test, p < 0.01*; Fig. 6). Therefore, the observations of a
temporal decrease of $Hg_{Anth}$ in the marine boundary layer of the NA (Sprovieri et al.,
2010; Soerensen et al., 2012; Weigelt et al., 2014) and 50%-loss of $Hg_{Anth}$
concentrations in the LSW over the last two decades are consistent. This means that
LSW formed in the 1990s' in the LS, and currently present in the ENAB, received more
$Hg_{Anth}$ from the atmosphere than the $LSW_{2014-2015}$ "vintage". These results contrast with



what can be deduced from the vertical profile of $HgT_{UNF}$ in the LS, where the Hg
regeneration in the water column is sufficient to account for the Hg increase between
the shallow LSW layer ($LSW_{2014-15}$) and the deep LSW layer ($LSW_{1987-94}$) (Cossa et al.,
in press). This discrepancy between these two deductions suggests that LSW, which are
present in the Eastern NA, is likely older (and more imprinted by legacy $Hg_{Anth}$) than
the LSW currently present in the LS.

Nonetheless, the decrease in $HgT_{UNF}$ in the NA is independently evidenced by the

current observations compared with data collected in 1993 and 1994, in both Subpolar
and Subtropical gyres, respectively (Mason et al., 1998; Cossa et al., 2004). In the
Subpolar Gyre, at that time, the Hg concentration in the LSWs layers ranged from 0.55
to 1.64 pmol $L^{-1}$. In the Subtropical Gyre, a multipeak pattern was also observed in 1994
in the Eastern Atlantic slope water column in the Celtic Sea (Cossa et al., 2004). The
shape of the Hg profiles exhibited the same peaks in the same water masses as the ones
observed in this study (i.e., SPMW and MW). However, $HgT_{UNF}$ concentration levels,
measured 20 years ago, were much higher varying mostly often from 0.3 pmol $L^{-1}$ in
sub-surface waters to more than 2.0 pmol $L^{-1}$ at depth. A decrease in HgT
concentrations over the last three decades supports the estimated decline in Hg
concentrations in subsurface waters of the NA estimated by models (e.g., Soerensen et
al., 2012).
*5.3.  Latitudinal transport of Hg*
The transport of $HgT_{UNF}$ per unit of water mass, calculated with Eq. (2) (Material and
Methods section), are given in Table 2. We also applied Eq. (2) separately to the upper
and lower limbs of the AMOC and computed the transports of $HgT_{UNF}$ per water mass



for the two limbs. The velocity fields across the Portugal-Greenland transect was
calculated using inverse model constrained by Doppler current profiler velocity
measurements (Zunino et al., this issue). The volume transport per SWT was computed
by combining this velocity fields with the results of the eOMP (García-Ibáñez et al., this
issue).

There is a northward $HgT_{UNF}$ transport within the upper limb of the AMOC (10.8

mmol s$^{-1}$), and a southward $HgT_{UNF}$ transport within the lower limb (7.3 mmol s$^{-1}$).
Most of the $HgT_{UNF}$ southward transport is due to IrSPMW and PIW displacements,
whereas $HgT_{UNF}$ northward transport is associated with $ENACW_{12}$ and $SPMW_8$
displacements. In addition, the mean (velocity-weighted) $HgT_{UNF}$ concentration of the
water advected northwards within the upper limb of the AMOC (0.58 pmol L$^{-1}$) is
higher than the one advected southwards within the lower limb of the AMOC (0.42
pmol L$^{-1}$). Thus, across the Portugal-Greenland transect, there is a net transport of 111
kmol yr$^{-1}$ from mid-latitudes to polar zones. To be able to estimate the Hg exchange
between Arctic Ocean and NA, the LS section has to be taken into account. The Hg
transport associated with the LC can be roughly estimated at 133 kmol yr$^{-1}$, using the
mean southward water transport of the shelf edge LC is at the Seal Island transect
(Hamilton Bank near Stas. 77 and 78) which is 7.5 Sv, according to Han et al. (2008)
and a mean HgT concentration of 0.56 pmol L$^{-1}$ (Cossa et al., in press). Thus, as a first
approximation, the net Hg exchange between the Arctic Ocean and NA would be an
Arctic loss of 22 kmol yr$^{-1}$, a smaller value than the estimate (130 kmol yr$^{-1}$) proposed
by the budget recently built by Soerensen et al. (2016).

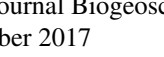



## 6.     Summary and conclusions

$HgT_{UNF}$ concentrations in the waters along the GEOTRACES-GA01 transect, which

crossed the NA from 40°N to 60°N (Portugal to Canada), ranged from 0.16 to 1.54

pmol $L^{-1}$, but with 97 % lower than 1.00 pmol $L^{-1}$ and a geometric mean of 0.51 pmol $L^{-1}$

$^{1}$ (n = 535). Below 100 m, most of $HgT_{UNF}$ (62-98%) is present as dissolved species (<

45 µm). $HgT_{UNF}$ concentrations increased eastwards and downwards, with the highest

$HgT_{UNF}$ concentrations found in the Subtropical Gyre, and especially within $NEADW_{L}$.

The relationship between $HgT_{F}$ and AOU reflects organic matter regeneration processes

on HgT mobilization and confirms a nutrient-like behavior for Hg in the NA. Using the

$HgT_{UNF}$ fraction unexplained by regeneration processes as a proxy for $Hg_{Anth}$, we

observed geographical and temporal trends in the $Hg_{Anth}$ in the NA. First, the highest

proportions of $Hg_{Anth}$ were found in the upper layer of the eastern NA ($ENACW_{12}$ and

$SPMW_{8}$). Secondly, there is an eastward increase within LSW, which suggests that Hg

incorporation in the downwelling waters of the LS has decreased over the last 20 years,

in parallel with the decrease of Hg concentrations in the NA troposphere. A net

northward Hg transfer of 111 kmol $yr^{-1}$ across the Portugal-Greenland transect results of

the AMOC. Taking into account the southern Hg export with the LC, the net Hg

exchange between the Arctic Ocean and NA would be an Arctic loss of 22 kmol $yr^{-1}$.

*Abbreviations*: AABW, Antarctic Bottom Water; AFS, atomic fluorescence
spectrometer; AMOC, Atlantic Meridional Oceanic Circulation; AOU, Apparent
Oxygen Utilization; CFCs, chlorofluorocarbons; CRM, certified reference material; DL,
detection limit; DSOW, Denmark Strait Overflow Water; DWBC, Deep Western
Boundary Current; EGC, Eastern Greenland Current; ENAB, Eastern North Atlantic
basin; ENACW, East North Atlantic Central Water; eOMP, extended Optimum
Multiparameter analysis; Hg, mercury; $HgT_{Anth}$, anthropogenic HgT; HgT, total
mercury; $HgT_{UNF}$, unfiltered HgT; $HgT_{F}$, filtered HgT; IAP, Iberian abyssal plain; IcB,



Iceland basin; IOC, International Oceanographic Commission; IrS, Irminger Sea;
ISOW, Iceland-Scotland Overflow Water; LC, Labrador Current; LS, Labrador Sea;
LSW, Labrador Sea Water; MW, Mediterranean Water; NA, North Atlantic Ocean;
NAC, North Atlantic Current; NADW, North Atlantic Deep Water; NEADW$_L$, Lower
North East Atlantic Deep Water ; PIW, Polar Intermediate Water; SPMW, Subpolar
Mode Water; SWT, Source Water Type; WGC, Western Greenland Current.

*Acknowledgments*: The first thanks are for J. Boutorh, M. Cheize, J.-L. Menzel and R.
Shelley who were in charge of the ultra-trace sampling organization; they are
commended for its successful achievement. Thanks are also due to other members of the
GEOVIDE team for participating to data acquisition: F. Alonso Pérez, R. Barkhouse, V.
Bouvier, P. Branellec, L. Carracedo Segade, M. Castrillejo, L. Contreira, N. Deniault, F.
Desprez de Gesincourt, L. Foliot, D. Fonseca Pereira, E. Grossteffan, P. Hamon, C.
Jeandel, C. Kermabon, F. Lacan, P. Le Bot, M. Le Goff, A. Lefebvre, S. Leizour, N.
Lemaitre, O. Menage, F. Planchon, A. Roukaerts, V. Sanial, R. Sauzède, and Y. Tang.
A special thank is also due to the R/V "*Pourquoi Pas?"* crew and Captain G. Ferrand,
and the DT INSU (E. de Saint Léger, F. Pérault) who organized the rosette
deployment/recovery processes. This research was founded by the French National
Research Agency (ANR-13-BS06-0014, ANR-12-PDOC-0025-01), the French National
Center for Scientific Research (CNRS-LEFE-CYBER), the LabexMER (ANR-10-LABX-
19), the Global Mercury Observation System (GMOS, N°265113 European Union
project), and the European Research Council (ERC-2010-StG-20091028). For this work
M.I. García-Ibáñez and F.F. Pérez were supported by the Spanish Ministry of Economy
and Competitiveness through the BOCATS (CTM2013-41048-P) project co-funded by
the Fondo Europeo de Desarrollo Regional 2014-2020 (FEDER).

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

F.: The GEOVIDE cruise in May-June 2014 revealed an intense MOC over a cold and
fresh subpolar North Atlantic. This issue.





**Figure caption**
**Figure 1.** Schematic view of the water circulation in the North Atlantic Ocean adapted
from García-Ibáñez et al. (2015) and Daniault et al. (2016). Red lines indicate the
circulation in surface, while blue lines indicated circulation at depth. Black lines
represent the GEOVIDE cruise transects (GEOTRACES-GA01). Main geographical
features, water masses and currents are indicated: Newfoundland (NFL), United
Kingdom (U.K.), United States of America (U.S.A.); Denmark Straight Overflow Water
(DSOW), Iceland-Scotland Overflow water (ISOW), Labrador Sea Water (LSW),
Lower North East Atlantic Deep water ($NEADW_L$), Mediterranean Water (MW), and
North Atlantic Deep Water (NADW), Deep Western Boundary Current (DWBC),
Eastern Greenland Current (EGC), Labrador Current (LC), North Atlantic Current
(NAC), Western Greenland Current (WGC).
**Figure 2**. Distribution of unfiltered total mercury ($HgT_{UNF}$) concentrations along the
GEOTRACES-GA01 transect. LS: Labrador Sea; IrS: Irminger Sea; IcB: Iceland basin;
ENABw: west part of Eastern North Atlantic basin; ENABe: east part of Eastern North
Atlantic basin; IAP: Iberian Abyssal Plain.
**Figure 3.** Mercury concentrations in filtered ($HgT_F$) *vs* unfiltered ($HgT_{UNF}$) samples (n
= 141) collected along the GEOTRACES-GA01 transect.
**Figure 4.** Total Hg in filtered samples ($HgT_F$) *vs* apparent oxygen utilization (AOU)
relationship along the GEOTRACES-GA01 transect.
**Figure 5.** Total Hg in unfiltered samples ($HgT_{UNF}$) *vs* apparent oxygen utilization
(AOU) relationship within the various Source Water Types.
**Figure 6.** Anthropogenic HgT ($HgT_{Anth}$) concentration distribution in the core of the
Labrador Sea Water (LSW) (S = 34.9, $\sigma_\theta$ = 27.74–27.82, 1200–2000 m) between the
Labrador Sea and the Eastern North Atlantic basin. $HgT_{Anth}$ values were obtained
according to the model by Lamborg et al. (2014). Young LSW corresponds to the
"2014-vintage" ($LSW_{2014}$) formed during winter 2013–2014. The insert shows the Hg
concentration decrease in the troposphere over the North Atlantic during the last 20
years according to Soerensen et al. (2012).



**Tables**

**Table 1**. Total Hg in unfiltered samples (HgT$_{UNF}$) *vs* apparent oxygen utilization (AOU) concentrations of each source water type (SWT), calculated according to eOMP (Eq. 1) (see also García-Ibáñez et al., this issue). ENACW$_{12}$: East North Atlantic Central Water of 12°C; SPMW$_8$ and SPMW$_7$: Subpolar Mode Waters of the Iceland basin of 7 and 8°C; IrSPMW: Subpolar Mode Water of the Irminger basin; LSW: Labrador Sea Water; MW: Mediterranean Water; ISOW: Iceland-Scotland Overflow Water; NEADW$_L$: lower North East Atlantic Deep Water; DSOW: Denmark Strait Overflow Water; PIW: Polar Intermediate Water; and SAIW$_6$: Subarctic Intermediate Water of 6°C.

| SWT | HgT$_{UNF}$ (pmol L$^{-1}$) | AOU (µmol L$^{-1}$) |
|---|---|---|
| ENACW$_{12}$ | 0.47 ± 0.01 | 13.4 ± 1.3 |
| SPMW$_8$ | 0.77 ± 0.03 | 112.8 ± 3.1 |
| SPMW$_7$ | 0.54 ± 0.03 | 65.4 ± 2.7 |
| IrSPMW | 0.26 ± 0.03 | 20.8 ± 2.7 |
| LSW | 0.46 ± 0.01 | 37.6 ± 1.0 |
| MW | 0.75 ± 0.04 | 84.7 ± 4.3 |
| ISOW | 0.59 ± 0.02 | 53.3 ± 2.2 |
| NEADW$_L$ | 1.02 ± 0.02 | 98.6 ± 2.0 |
| DSOW | 0.43 ± 0.03 | 32.0 ± 3.0 |
| PIW | 0.73 ± 0.11 | 77.3 ± 10.8 |
| SAIW$_6$ | 0.45 ± 0.03 | -12.7 ± 3.0 |



**Table 2**. Water and total Hg in unfiltered samples (HgT$_{UNF}$) transport by the upper and
lower limbs of the Atlantic Meridional Overturning Circulation. Positive (negative)
transports correspond to northward (southward) flow.

| SWT | Entire water column | | Upper limb | | Lower limb | |
|---|---|---|---|---|---|---|
| | Water transport (Sv) | HgT$_{UNF}$ transport (mmol s$^{-1}$) | Water transport (Sv) | HgT$_{UNF}$ transport (mmol s$^{-1}$) | Water transport (Sv) | HgT$_{UNF}$ transport (mmol s$^{-1}$) |
| ENACW$_{12}$ | 9.6 | 4.65 | 9.6 | 4.65 | 0.0 | 0.00 |
| SPMW$_8$ | 4.1 | 3.28 | 3.7 | 2.92 | 0.5 | 0.36 |
| SPMW$_7$ | 3.2 | 1.75 | 1.8 | 0.99 | 1.4 | 0.76 |
| IrSPMW | -10.1 | -2.73 | -0.8 | -0.22 | -9.3 | -2.51 |
| LSW | 1.5 | 0.70 | 2.7 | 1.31 | -1.3 | -0.60 |
| MW | 0.7 | 0.56 | 0.6 | 0.50 | 0.1 | 0.06 |
| ISOW | 0.9 | 0.55 | 1.2 | 0.76 | -0.3 | -0.21 |
| NEADW$_L$ | 0.3 | 0.33 | 0.0 | 0.00 | 0.3 | 0.33 |
| DSOW | -2.2 | -0.98 | -0.1 | -0.06 | -2.1 | -0.91 |
| PIW | -4.8 | -3.59 | 0.0 | 0.00 | -4.8 | -3.59 |
| SAIW$_6$ | -2.1 | -0.99 | 0.0 | 0.00 | -2.1 | -0.99 |
| TOTAL | 1.1 | 3.52 | 18.7 | 10.84 | -17.7 | -7.32 |






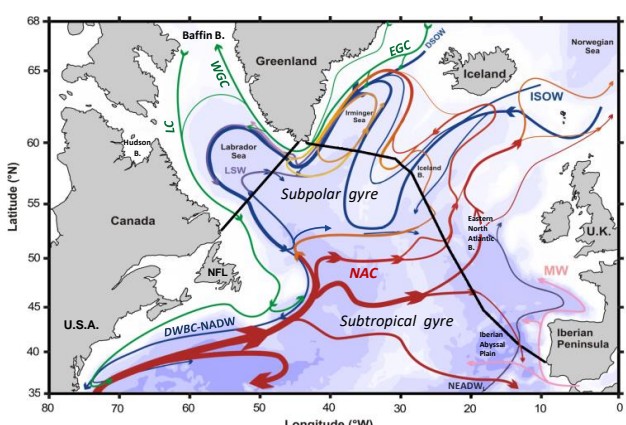

Fig. 1

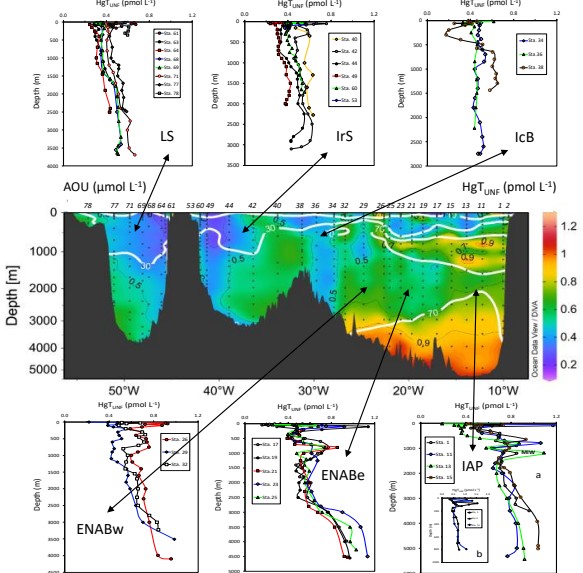

Fig. 2




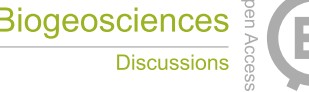


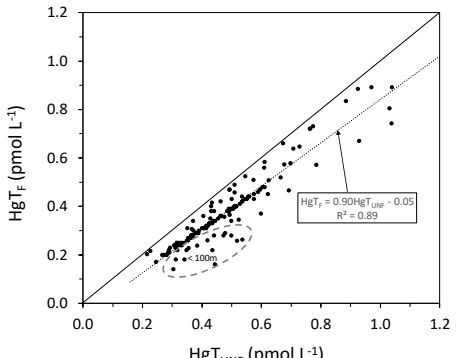

Fig. 3



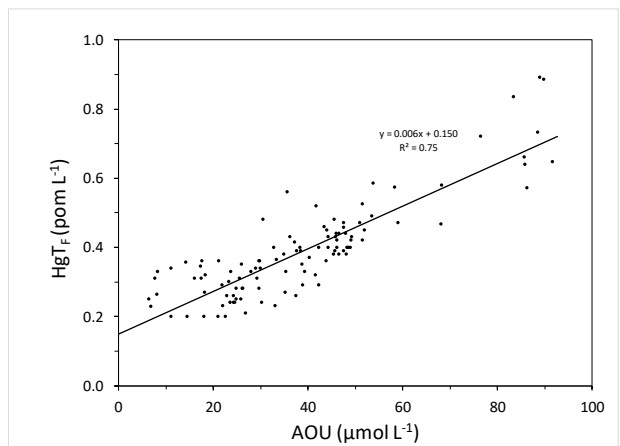

Fig. 4








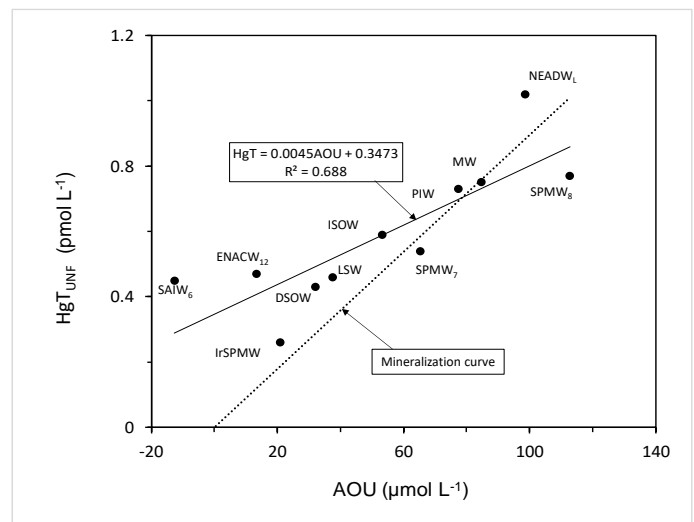

Fig. 5


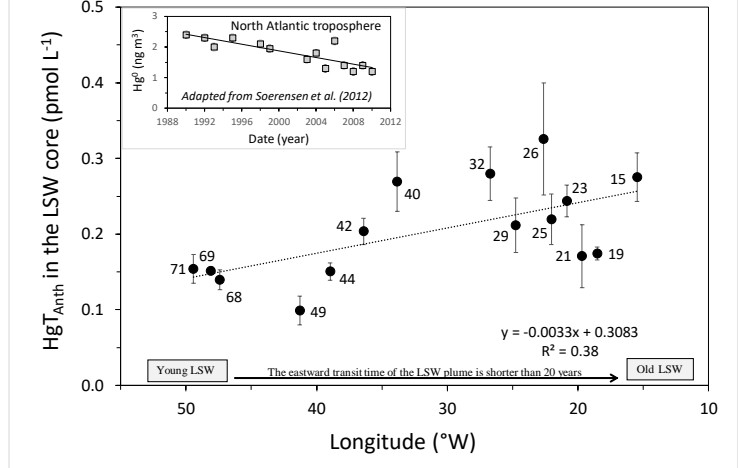

Fig. 6

