# Peer review of "Mercury distribution and transport in the North Atlantic Ocean along the GEOTRACES-GA01 transect"

_Biogeosciences, 2017_

## Short Comment (SC1) · 16 Dec 2017

Great paper describing concentrations and trends from a critical stretch of ocean. The incorporation of the eOMP is particularly interesting. I was left wishing the authors had done a bit more with this information. For example, could they have compared the eOMP reconstructed end-members to the age of those water masses across the section? Something like this is done in Figure 6 for the LSW, but I was curious to see it for all the water masses.

The difference between filtered and unfiltered was interesting to see, and I was wondering if there was any particle mass data generated by the science team such that a

Kd could be generated?

Line 20, why the geometric mean rather than the arithmetic mean? Lines 38-39, net flow toward the poles? Can the authors comment on how this can be? Evasion at high latitude is the balance? Line 55, "leading to a doubling"...some of the papers cited suggested a higher impact, more like factor 3...perhaps put a range in here. Line 95, "several water masses are stacked up from surface to bottom..." sounds a little awkward. Perhaps change to "several water masses can be identified. Listing them from top to bottom..." Line 104, change to read "...often notated with a temperature as a subscript..." Line 119, "LSW has been variably produced in the past fifty years..." Given the apparent importance of LSW and AMOC as a sink for Hg, is the variability in LSW production reflected in atmospheric concentrations? For example, at times of weak LSW formation, was the atmospheric Hg concentration a bit higher than times when LSW formation was strong? Line 292, a concentration of 63 pmole/L is mentioned...typo? Line 323, change "amplitude" to "magnitude". Later, the phrase "the position of the upper peaks suggests a relation with the abundance of phytoplankton." It was unclear to me what that was based on...abundance of chlorophyll? Line 356, change "entails the existence of relationships" to "results in a relationship" Line 369-371, comparing the measured to the predicted values is worthwhile, but I was unclear what the implications of the correlation coefficient were. The fit should be fairly good since the measured were used to generate the predicted values...is the point that the fit is not 100%? The point of the remineralization line in Figure 5 was a little confusing to me. Is the essential point that the real data vs. AOU have an intercept as opposed to going through zero? The only difference between the two lines is that one is forced through the origin and the other is not...so, the discussion about the water masses on the low AOU end being above the line felt like a "self-fulfilled prophecy."

---

## Referee Comment (RC1) · Anonymous Referee #1 · 21 Dec 2017

Review of Cossa et al. "Mercury distribution and transport in the North Atlantic Ocean along the GEOTRACES-G01 transect"

General comments: This manuscript presents a new set of high quality measurements in an oceanographically important region. These data add to our understanding of variability in oceanic Hg concentrations across different water masses. In general, I find the work to be high quality. I have a number of suggestions for improvements/corrections that are detailed in the specific comments. Most importantly, the authors make a number of assumptions about processes and mixing in their eOMP, leading to a statistical estimate of source water contributions, which are later used to infer anthropogenic in-

fluences and fluxes across major regions. They contrast their approach with "modeling" estimates throughout the manuscript but their approach is also a modeling estimate (albeit a statistical one) and based on a measurement snapshot and should be acknowledged as such. I would like more information on the specifics of how this analysis was performed, details of the equations and specific mixing assumptions, and implications of any assumptions for results. For example, the authors ignore the influences of Hg losses through evasion etc. throughout much of their discussion and analysis and I wonder about this implications of doing this. The conclusions about anthropogenic Hg changes seem tenuous to me and should be more clearly justified in the revised paper.

Specific comments:

Abstract, Line 18: What about the other 3 percent? Why not just present the full range and some measure of central tendency?

Line 21: The particulate fraction in this work seems higher to me than in other ocean regions and the authors may want to comment on this.

Line 43: Paper by Alex Poulain's work in NGS contradicts this statement – might want to acknowledge and caveat.

Line 46: "ocean-atmosphere exchange" is important for the global biogeochemical cycle of Hg but I don't think one can say that it "dominates"

Line 48: Neither of the papers cited is about the atmosphere.

Line 49: Seems important to mention Hg redox cycling and losses through evasion here.

Line 53: There are a number of other estimates of anthropogenic Hg impacts on the ocean that should be acknowledged.

Line 56: As per comment above – I don't think doubling is an appropriate representation of the consensus in the literature or lack thereof. This is a relatively controversial topic

and not directly relevant to the work presented here so the authors may simply want to omit these numerical statements. The review by Amos et al. (2015) cited later in the manuscript also explicitly discusses the range in oceanic enrichment suggested by recent work and agreement of such ranges with measured concentrations.

Lines 60-63: This seems like an odd rationalization for this work since most policy makers probably don't even know what "ocean water mass" means and certainly this is not at the forefront of discussions. I would prefer to see scientific objectives and goals since this is not intended as a policy friendly paper.

Lines 65-66: Grammatical problems. I don't think "solubility pump" is appropriate here since Hg solubility is not inversely proportional to temperature in the same way as $CO_2$ – why not just make a direct statement about thermohaline circulation/advective transport since this is what they mean.

Lines 71-72: I don't think it is only GEOTRACES that has produced these results. Please rephrase.

Lines 79-84: Would be helpful to have an overarching objective for the work, i.e., how biogeochemical variability in the oceans affects total Hg concentrations. These reads like a list of unrelated tasks.

Line 131: I think this should be "...long time since ventilation" rather than "negligible atmospheric exchange"

Lines 132-133: I think the authors are simply trying to say some Hg is transported advectively with ocean circulation rather than it "follows CFCs" – CFCs are simply unreactive tracers for circulation so it is misleading to imply they interact with Hg at all and that could be interpreted from the current phrasing.

Line 137: I think the phrasing here is also problematic because CFC concentrations are not fixed and depend on water mass mixing, time since ventilation and concentration of CFCs in the atmosphere at the last ventilation period. I would therefore recommend

that this refer to a "past" condition/measurement from a given region.

Line 138: Again – be specific and refer directly to advective transport with seawater and particles. All of this is unnecessarily confusing.

Lines 177-182 – Some of this seems unnecessary – these are standard methods.

Line 212-218: I would like more information on the eOMP, the residual term from the analysis, and the implications of assumptions/details like omitting water masses above 75 m depth. Equation one seems to imply the eOMP is solved simply as a mixing model – which makes sense for DOC given its long lifetime but this is not capturing some major characteristics of Hg cycling such as evasion, diffusion in the water column, uptake by food webs and particle settling. It seems like this might work well under specific conditions where these other processes have a limited influence on total Hg distribution but do think this can be assumed to be always the case.

Line 256: If it is a skewed lognormal distribution shouldn't the data be lognormally transformed?

Line 270: Also variability in biogeochemistry such as DOM affecting pool of Hg available for reduction.

Line 272: Why exclude the upper 100 m?

Line 351: Need to mention Hg evasion again.

Line 353: Also diffusion and eckman pumping.

Line 362: Not that AOU is also affected by lateral transport processes.

Line 369: AOU not from in situ measurements of oxygen and temperature or AOU for source waters estimated? Please clarify.

Line 372-374 – Almost no English in these sentences – I can't remember all the acronyms used so it would improve readability to get rid of some of them.

Line 382: Why would AABW be enriched in total Hg?

Line 402: We already know this. Please rephrase to acknowledge prior work.

Line 403: Reference and inference here is problematic. The DeSimone paper simply refers to primary anthropogenic releases for a given time period not all anthropogenic Hg contributions to deposition and there is variability across models in these estimates and even how you define the human fraction. This seems irrelevant to me to this paper so I suggest just deleting the problematic statement.

Line 408: Don't you mean particle scavenging not OM regeneration? OM regeneration is associated with export.

Line 415-416: The analytical uncertainty around the early numbers is very large. Suggest acknowledging this here as well.

Line 420. I am not convinced by the proposed relationship with remineralized phosphate and anthropogenic Hg proposed in this paper. Suggest rephrasing to acknowledge this is a proposed relationship not an established tracer for anthropogenic Hg.

Line 434: Subsurface ocean concentrations and their decreases is an assumption in Soerensen et al. 2012 based on observations as a boundary condition in the model to test the influence on atmospheric trends. It is misrepresented by this statement.

Line 480-481: Soerensen et al. 2016 is a synthesis of observations and calculation from established measurements. This work considered all available measurements and represents a specific time period. This study represents a snapshot of measurements and a different time period. To imply that they should be directly compared is incorrect and these differences in these values acknowledged. I would call the value presented in this work a different "model" estimate. Both are derived by multiplying concentrations by flows.

Lines 491-493: I think the paper needs to make a more compelling case for this being reasonable.

Technical Corrections:

Line 57: I don't think anthropization is a word.

Line 59: I think it is preferable to use gender neutral language.

Line 147. Delete "etc." – not helpful. L ine 281: High is very subjective term. Suggest being more precise.

---

## Referee Comment (RC2) · Anonymous Referee #2 · 4 Jan 2018

General comments:

This study presents an impressive set of high resolution THg measurements in the North Atlantic Ocean (NA), along a transect from Lisbon (Portugal) to the Labrador Coast (Canada) as part of the GEOTRACES programme. The NA, where water masses mixing and deep water formation occur, is a location of interest to assess inputs of Hg to ocean water. This study also uses an interesting approach, extended Optimum Multiparamater (eOMP) analysis, to characterize THgUNF. concentrations relative to source water masses and to assess the anthropogenic Hg contribution to water masses.

[Figure]

While there is a fully detailed oceanography context on the formation of the water masses, a more complete description of the biogeochemistry of Hg in oceans, especially Hg evasion and deposition at the air-water interface which is identified as an important mechanism driving oceans THg concentrations, might be useful to the reader.

Description of the Hg vertical patterns in the different oceanographic regions could be condensed focusing on stations of interest where influence of water masses or specific oceanographic conditions are observed.

Are the mean THgUNF concentrations statistically different in LS, IrS and IcB??

Since SWTs are characterized by potential temp, salinity and nutrients, it might be interesting to see the plot of these parameters in SI. Please specify the nutrients used for the characterization of the SWTs.

Adding a salinity distribution plot with the water masses or SWTs superimposed on figure 2 might help the reader through the results and discussion sections.

The authors conclude atmospheric deposition is driving THg concentrations but they do not provide strong arguments (e.g. deposition rates, GEM concentrations etc.) to support this statement.

Specific comments:

Line186: How was the Hg-free seawater sample used as blank solutions prepared? Did you use the same blank solution during the cruise?

Line 221: specify which macro-nutrients were used for SWT characterization

Line 267: suggest "in addition, lowest and highest . . ."

Line 268: Rather speculative statement. Do you have additional data, e.g., wind forcing, other gases such as $CO_2$, flux, GEM in atmosphere or THg in particulate matter or high OM content at stations where elevated THg concentrations were recorded to

support this statement?

Line 292: Typo error "0.63" instead of "63"

Line 298: Is Hg evasion and/or PM content in this basin of greater importance than in the other areas so that they can explain the variations in surface THg concentrations (observed at Stn 38 only)?

Line 393: typo error? "SPMW7" instead "SPMW8"?

Lines 401-404: The authors explain the departure of estimated THg concentrations from the THgUNF vs AOU line, thus Hg enrichment, in SAIW6 and ENACW12 by atmospheric deposition. They state further that atmospheric deposition is a significant source of Hg to NA. However, THgUNF in these SWTs are relatively low (see Table 1). Hard to reach such a conclusion based on THgUNF concentrations in SAIW6 and ENACW12. While these results suggest that a process other than AOU, most probably atmospheric deposition is the main source of THgUNF in surface water (Upper limb of AMOC), it is of minor contribution compared to ocean circulation in NA water column. As mentioned in lines 406 – 408, OM regeneration and hydrological processes are the main factors controlling THg in NA waters. When OM regeneration not occurring THg concentrations are low stressing the lesser importance of other sources i.e., atmospheric deposition. I suggest to replace the term "significant" by "dominant".

Figure 4: is the relationship linear or should a non-linear relationship be investigated?

Technical corrections:

Line 252: Numbering error, results is section 4 instead of 5? Subsections should be corrected accordingly

Line 324: "...abundance of phytoplankton, whereas the position of the lower peaks, which is close to the maximum of Apparent Oxygen Utilization (AOU) that rose above 70 $\mu$mol L-1 (Fig. 2), suggests a dependence ...", report r-values for relationships and/or show plot of Chla / fluorescence distribution in SI

Line 342: add a salinity plot in SI

---

## Author Response (AR1)

**Mercury distribution and transport in the North Atlantic Ocean along the GEOTRACES-GA01 transect**

| | |
|---|---|
| Daniel Cossa[1] | dcossa@ifremer.fr |
| Lars-Eric Heimbürger[2] | lars-eric.heimburger@mio.osupytheas.fr |
| Fiz F. Pérez[3] | fiz.perez@iim.csic.es |
| Maribel I. García-Ibáñez[4] | maribel.garcia-ibanez@uni.no |
| Jeroen E. Sonke[5] | jeroen.sonke@get.omp.eu |
| Hélène Planquette[6] | helene.Planquette@univ-brest.fr |
| Pascale Lherminier[7] | pascale.lherminier@ifremer.fr |
| Julia Boutorh[6] | julia.boutorh@univ-brest.fr |
| Marie Cheize[6] | marie.cheize@univ-brest.fr |
| Jan Lucas Menzel Barraqueta[8] | jmenzel@geomar.de |
| Rachel Shelley[6] | rachel.shelley@univ-brest.fr |
| Géraldine Sarthou[6] | Geraldine.Sarthou@univ-brest.fr |

Daniel Cossa[1]

Lars-Eric Heimbürger[2]

Fiz F. Pérez[3]

Maribel I. García-Ibáñez[4,3]

Jeroen E. Sonke[5]

Hélène Planquette[6]

Pascale Lherminier[7]

Julia Boutorh[6]

Marie Cheize[6]

Jan Lucas Menzel Barraqueta[8]

Rachel Shelley[6]

Géraldine Sarthou[6]

*[1]ISTerre, Université Grenoble Alpes, CS 40700, F-38058 Grenoble Cedex 9, France*
*[2]Aix Marseille Université, CNRS/INSU, Université de Toulon, IRD, Mediterranean Institute of Oceanography (MIO) UM 110, Marseille, France*
*[3]Instituto de Investigaciones Marinas, CSIC, Eduardo Cabello 6, E-36208 Vigo, Spain*
*[4]Uni Research Climate, Bjerknes Centre for Climate Research, Bergen 5008, Norway*
*[5]CNRS, GET-OMP, 14 Ave. E. Belin, F-31240 Toulouse, France*
*[6]LEMAR, Université de Bretagne Occidentale, F-29280 Plouzané, France*
*[7]IFREMER, Brittany Center, LPO, BP 70, F-29280 Plouzané, France*
*[8]GEOMAR, Helmholtz Centre for Ocean Research, D-24148 Kiel, Germany*